# "There's No Discrimination, These Are Just the Rules of the Game": Russian Scholars' Perception of the Research Writing and Publication Process in English

Irina Shchemeleva 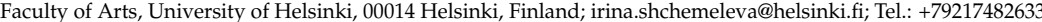

Faculty of Arts, University of Helsinki, 00014 Helsinki, Finland; irina.shchemeleva@helsinki.fi; Tel.: +79217482633

**Abstract:** This paper reports on the study of multilingual speakers' perception of their research writing practices in English and in their local language—Russian—and the publication process in English. It is based on interviews with 18 scholars from social sciences and humanities working in a leading university in Russia. The study discusses social factors influencing multilingual scholars' choice of languages as well as their personal motivation to choose English as the main language of publication. Special attention is given to their attitude to proofreading as part of the publication process. The interview results suggest that, from the participants' perspective, the benefits they gain by publishing research in English seem to outweigh costs they experience in the process of writing and publishing. The study contributes to the on-going debate about the position of multilingual scholars in the competition to publish in top-rated journals, suggesting that the traditional doctrine of linguistic injustice, from the participants' point-of-view, does not seem to be relevant for every multilingual scholar.

**Keywords:** multilingual scholars; research writing in English; writing practices; publication process

## 1. Introduction

One of the key strategic priorities of contemporary universities has become internationalization, generally understood as "the integration of an international or intercultural dimension into the tripartite mission of teaching, research, and service functions of higher education" [1], (p. 1). A process closely related to internationalization is a wide use of English in academia, or its Anglicization [2–4]. To build a strong reputation as well as to attract international students and researchers, universities around the globe design English as a medium of instruction programs and open international research laboratories. This means that both within and outside of Anglophone contexts, multilingual scholars use English as the language of teaching instruction and research.

Another characteristic of contemporary academia is a focus on accountability [5,6], which is manifested in the implementation of quality assessment schemes [7]. Research shows that, in different geo-linguistic contexts, scholars' promotions, career opportunities, and rewards depend on their publication efficiency [8–10]. Moreover, research quality and productivity are measured not only in the number of papers scholars write, but also in the number of citations they receive. As a consequence, scholars are encouraged to publish in top journals with high citation indexes, which are English-medium journals in the majority of cases. For non-Anglophone contexts, it means that publication in English is favoured compared to publication in local languages [9,11].

In the last two decades, there has been a wealth of research devoted to the studies of multilingual scholars' academic writing practices conducted in various regions and contexts (for a review, see Reference [12], (p. 2)). These studies, having different objectives, have much in common. They describe policies at governmental and institutional levels aimed at increasing the number of publications in top-ranked journals [13–17]. They mention the principle 'publish in top-ranked journals or perish' [10,15] that very often equals 'publish

in English or perish' [8,18–20], and they show that English as a language of publication replaces local languages [21,22].

Almost a decade ago, Hyland [23] made the following observation:

> The expression "publish or perish" has probably never been as cruelly applicable as it is today. Universities in many countries now require their staff to present at international conferences and, more crucially, publish in major, high impact, peer-reviewed Anglophone journals as a pre-requisite for tenure, promotion, and career advancement. Academics all over the world are increasingly less likely to publish in their own languages and to find their English language publications cited more often (p. 37).

Since then, the competitiveness of universities, especially for success in international university rankings, has increased. This means that the pressure for scholars around the world to publish in high-ranked journals in English has intensified [6,18,24].

Curry and Lillis [10] identify two key tensions that appear in many non-Anglophone countries as a result of the 'publish in English or perish' policy. The first is that scholars are discouraged to contribute to journals that are not highly ranked, including journals in local languages, and the second concern is that local and regional contexts become deprived of important knowledge (p. 3). Other researchers warn that policies to reward publications in top journals in English work against scholars who are committed to sharing knowledge with the local community in the local language [25,26]. These concerns are supported by many studies of research writing practices by multilingual speakers in non-Anglophone countries [5,18,27].

Research writing and a publication process in English is treated by some scholars as a game with its own rules and strategies, which academics need to learn, follow, and skilfully practice [28]. Casanave, for example, shows that academic writing is a serious "game-like social and political as well as discoursal practice" [28], (p. 1). In the literature, a common attitude toward multilingual scholars' practices of research writing in English is well formulated by Curry and Lillis [10] who, using Bourdieu's [29] metaphor of fields and referring to the results of the studies by Canagarajah [30] and Mweru [31], argue that multilingual scholars "enter the academic publishing 'game' on a 'field' that is not level, both in terms of access to English as well as to other social and material resources (research and travel funding, library access) needed for publication" [10], (p. 5).

Flowerdew [32] identified the main burdens multilingual scholars face: restricted "facility of expression," a "less rich vocabulary," a "simple style," difficulty "to make claims for their research with the appropriate amount of force." He also mentioned that their writing could be influenced by their L1 (p. 243). Since then, the idea of the 'linguistic injustice' that multilingual scholars experience in the process of writing and publishing their research has been developed in many studies reporting that the scholars feel inequality compared to native speakers [33–39]. Even in the titles of publications, the authors stress difficulties [40,41] or treat writing research in L2 as a 'burden' [42]. The studies were conducted in different regions and included speakers of various L1, such as scholars from Argentina [43], Spain [9,36], Poland [26], and China [44]. The shared argument is that multilingual authors are disadvantaged when compared to L1 authors because they have a "dual burden in writing in English that considerably increases their workloads and generates anxiety and dissatisfaction" [12], (p. 6). The studies also report that reviewers and editors favour native English speakers [45]. Flowerdew [35] even uses the term "stigmatized" to describe how multilingual writers are treated by journal editors and reviewers.

Recently, however, several scholars have questioned the ideas that multilingual scholars have the additional burden and, in general, experience linguistic inequality. Hyland [46], recognizing that multilingual scholars face challenges getting their research published, especially in top-rated journals, argues that such difficulties have very little to do with the Native vs. Non-native divide. He maintains that academic literacy is a new competence for any scholar irrespective of their L1, and publishing in English may be equally difficult for both L1 and multilingual speakers. This idea is shared by other researchers [47,48]

who suggest that expertise in academic writing and publishing comes with practice. However, the difficulties in writing for publication may never disappear even for senior L1 researchers. Casanave [49], for example, believes that writing for publication does not always get easier. One of the arguments she provides is the increasing difficulty in keeping up with the rapid growth of information in scholarly fields (p. 144).

Another argument for the idea that multilingual speakers are not in a disadvantaged position is that language is not the main factor for the acceptance or the rejection of a paper [49]. The publication process is influenced by the formal training the scholars get, by the level of their experience in research writing, by the local academic culture they write their texts in, and by the collaboration they are involved in [46]. With regard to language use, studies indicate that such features as study design and paper organization may be more important than linguistic errors for the reviewers and editors [50]. Critical reviewer comments relate more to methodology and content than to language [48]. In some recent research, the authors have found that non-canonical linguistic expressions are accepted in published papers [51–53]. This suggests that some reviewers and journal editors are becoming more tolerant to the variability in language use by multilingual speakers of English.

There are also qualitative studies of multilingual scholars' perceptions of their writing practices that show the benefits scholars gain when they choose English as the language of publication. Martín et al. [40] in their investigation of research writing practices of Spanish medical researchers found that the most common motivation to publish in English was to "communicate the results of their research to the international scientific community" (p. 60). It is interesting that the next two reasons chosen by the participants were the desire "to be widely recognized by the community" and "to get cited more frequently" (p. 60), which can also be considered as personal benefits the scholars gain by publishing in English. In McDowell and Liardét's study [54] of Japanese scholars' experience with research writing in English, the participants named the advantages of collaboration and the possibility to disseminate information globally (p. 152) as the main benefits from publishing in English. The study found that the benefits in writing and publishing in English outweigh the burdens. In general, the studies questioning the application of the concept of 'linguistic injustice' to multilingual scholars, suggest that we may be witnessing some changes in attitude happening in academia today.

In this study, research writing is viewed as a social practice [55,56] influenced by social contexts in which it is produced. It is formed in the process of "working within social contexts and contending with the power relations of these contexts" [10], (p. 4). I approach multilingual scholars' writing as embedded in wider social and institutional context. In this respect, it is imperative to learn what motivates multilingual writers who do their research at the beginning of the third decade of the 21st century, to study the role of gatekeepers in the process of writing and publishing, and to discover multilingual scholars' perception of research writing and publishing in English. These issues have been addressed in recent publications. Some studies focused on the impact of national and institutional policies about publishing on multilingual scholars' writing practices [13–15,57,58]. Others explored the interventions of gatekeepers and 'literacy brokers' in the process of text production [34,48,59–61]. While these studies were conducted in various geographic locations, none of them examined Russian scholars' perceptions of research writing and their publication practices. This is of vital interest since Russian universities have put forth considerable effort to be ranked highly among international universities, which has resulted in a significant increase in the number of international students (Russia holds the eighth place in the number of international students in 2018–2019 [62]) and in scholarly publication output in international journals [63].

This study is based on the interview with 18 scholars from social sciences and humanities working in a leading university in Russia. They are middle-career researchers who are actively involved in national and international research collaboration and who are

successful in publishing their research in English. The following research questions guided the study.

(1)   What are multilingual scholars' publication practices: which languages they use and what influences their choice?
(2)   What is multilingual scholars' perception of English as the language of publication and the publication process in English?

From the critical language policy perspective, I study multilingual scholars' language practices, which Spolsky [64] defines as "observable behaviours and choices—what people actually do" and which are influenced by "beliefs", i.e., "values and statuses assigned to named languages" and "management" or "the explicit and observable effort by someone or some group that has or claims authority over the participants in the domain to modify their practices or beliefs" (p. 4). I aim to see how multilingual authors, under the influence of national and institutional policies, choose languages for research writing, and how journals influence the research writing practices by imposing the standards regarding English and advising to use proofreading services. I am also interested in learning if contemporary multilingual scholars see themselves in a disadvantaged position with regard to English compared to L1 speakers of English.

The study is context-oriented. It is 'situated' in one university, but it allows us to draw conclusions on tendencies regarding research writing and publishing in English and in Russian taking place in Russian universities that strive to be internationally recognised.

## 2. Materials and Methods

### 2.1. Context of the Study

Russian is the eighth most widely spoken world language [65]. In the USSR, it was the lingua franca of academia. Practically in all disciplinary fields, all research was published in Russian. The published works went through a rigorous review process that guaranteed quality. This situation started to change after the collapse of the USSR when many new research journals appeared. It made the publication process easier and faster.

Since the beginning of the 21st century, when internationalization emerged on the strategic agendas in many universities all over the world [4,66], leading Russian universities have undergone significant changes. They invested in developing better research opportunities and designing English-taught courses and programmes. The English language as the language of academia has gradually started to be used either parallel to Russian or instead of Russian, at least in the universities that aim at building a high international reputation.

To support the leading universities in the process of globalization and internationalization, at the state level, programmes that aimed at building universities' global reputation and enhancing their international visibility have been introduced. An example of such an initiative is the Russian academic excellence project launched in 2013 with the goal to "maximize the competitive position of a group of leading Russian universities in the global research and education market" [67]. One of the indicators of the universities participating in the project is their research performance measured in the number of publication and citation counts in Web of Science and Scopus. This has resulted in a dramatic increase in the number of publications in top journals by the researchers from the participating universities [68].

The setting of the study is one of the most competitive research-led universities in Russia that participates in the Russian academic excellence project. The university can be called an 'internationally-focused university' in the terminology of Foskett [69], (p. 44). It puts internationalization high on its agenda both in relation to education and research.

The university offers a range of degrees taught in English. The Russian-taught programmes also have courses taught in English. The university has strict requirements in terms of English for students and for faculty members. Their level of English should be high enough to study/teach courses and to use in professional communication, both oral and written.

Regarding the internationalization of research, the university attracts leading researchers from the international market and creates good conditions for research collaboration by opening international research centres and laboratories. Since 2005, the university has also been applying an academic bonus merit system that stimulates researchers to publish in top-ranked journals in their fields. There are different categories of bonuses, including bonuses for a publication in an international peer-reviewed academic journal and bonuses for academic success and contributions to the university's academic reputation. Depending on the quality of publication, the researcher receives monthly bonuses. Originally, top journals published in Russian and indexed in Web of Science and Scopus were included in the list of eligible journals, but, from 2021, these journals will be excluded, which means that no publication in Russian will be counted. The university provides support in research writing through the Academic Writing Centre that was opened 10 years ago. The centre organizes offline and online courses and workshops in academic writing skills that are designed to meet the requirements of different groups of scholars. They range in topics and are aimed at speakers with various levels of English proficiency. The Academic Writing Centre also holds individual consultations with professional proofreaders and editors.

The Russian academic excellence project and the academic bonus system described above are examples of "mechanisms" (in the terminology of Shohamy [70]) that "are used as means for affecting, creating, and perpetuating de facto language policies" (p. 52), i.e., the means that shape scholars' writing practices. The academic bonus system, for example, not only financially stimulates publishing in a certain language, which is English in the majority of cases, but also clearly sets preferences for a particular genre—the research article. A similar situation is observed in other countries (Tusting's [71] conclusions of the impact of the UK national research assessment process on scholars' writing practices).

## 2.2. Data Collection

This paper is part of a larger study of research writing practices in English of multilingual scholars representing a broad domain of disciplines in social sciences and humanities [72]. To select the participants for the study, the following criteria were applied: the L1 of the participant, departmental affiliation, the number of publications in English, and the career stage.

Although the university where the research was conducted had faculty members with various L1s, I decided to invite only scholars whose L1 was Russian so that the sample was homogeneous not only in terms of L1 of the participants, but also in terms of the academic culture they internalized. It was important that the participants acculturated in similar social and educational contexts. For the study, the Departments of Political Science, Sociology, Management, Public Administration, and History were chosen. It was not important to have equal representatives of each department. Rather the aim was to have representatives from a range of disciplines from social science and humanities.

For the purposes of the study, it was vital to have the participants who had experienced publishing in English. I searched departmental websites to find the researchers who had at least five publications in reputable journals in English. I assumed this number was enough to guarantee that they were familiar with the publications process in English.

Scholars' career stage as a criterion was chosen because many studies showed that it could have a significant impact on scholarly writing practices [24,73]. Based on Swales' dichotomy of junior vs. experienced researchers [74], middle-career researchers were chosen because, first, they seem to be active in research and, second, they started their professional careers at a time when English had already established itself as the major language of publication. This means that, from the earlier stages of their professional life, they were very likely to read papers in English and to choose English as the main language of publication.

At the initial stage, a case study with representatives from the political science department was conducted [72]. Then, I refined the interview protocol and invited representatives from other departments to participate. The aim was to have 20 participants. However,

only 18 replied to the request to take part in the interview. The participants represent all the departments that were chosen: five scholars are from the Departments of Political Science (PolSc), five scholars from the Department of Sociology (Soc), four scholars from the Department of History, three scholars from the Department of Management (Man), and one scholar is from the Department of Public Administration (PubAd). In the interviews, it turned out that many participants viewed themselves as multidisciplinary researchers. Some participants affiliated themselves with disciplines outside the disciplinary field of their department (for example, a participant from the department of sociology named cultural studies and the anthropology of art as the main field of research, or a scholar from the department of history affiliated himself, besides history, with communication, art history, and literary studies), but all of them belong to the broad field of social sciences and humanities. Hereafter, I use the departmental affiliation to refer to the participants, e.g., Soc1, Soc2, and Soc3.

The age of the participants ranges from 31 to 46 years. They all hold PhDs: 13 scholars received their degree from a Russian university, three scholars received their degree from an international university, and two scholars have two degrees from a Russian and an international university. According to the information presented on the website, in recent years, the average number of papers in English authored by each participant is nine. The average number of papers published in Russian is four.

With each participant, face-to-face semi-structured interviews in Russian were conducted. The excerpts that are included into the paper, were translated by the author. This paper reports on the part of the interview devoted to the use of English and Russian in the professional sphere and to the interviewees' perceptions of the publication process in English.

To ensure the validity of the results, an interview protocol was designed (Appendix A). Deviations from the protocol were allowed so that the issues emerged in the interview could be discussed at length. First, general questions about the participants' educational, linguistic, and professional background and their disciplinary affiliation were asked. Then the interviewees were asked a series of questions aimed to uncover which languages they used in the professional sphere and what influenced their choice.

Another set of questions was designed to elicit the discussion of the participants' research writing practices. They focused on the rationale for choosing English as the language of publication and the factors that determine their choice. The participants were also asked about the difficulties they experienced when they wrote research texts in English.

To find out the participants' views on the publication process in English questions about difficulties they face in the process were asked. To learn whether (and how), from the participants' point of view, the fact that they are non-native speakers of English influences the publication process in order to elicit the discussion of language-related issues in this process, I asked questions about the role of linguistic issues in the rejection/acceptance of papers. We also discussed their attitudes to proofreading and their experience of proofreading. The discussion of proofreading was included in the interview because this is what distinguishes multilingual scholars writing in English from their L1 peers. As is known, many top journals advise non-native speakers to have their texts proofread. In the interviews, the participants were also asked if they felt disadvantaged while getting their papers published compared to native speakers of English.

*2.3. Analysis of Interview Data*

For the data analysis, all the interview transcripts were read and coded independently by two researchers: the author of the paper and a scholar holding a PhD in linguistics whose interests include multilingual scholars' research publication practices. We combined two approaches to coding discussed by Stemler [75]: the codes that were a priori created based on the interview protocol questions and codes that emerged in the transcripts were used. At the initial stage, a close reading of the transcripts to identify themes was done. Then any discrepancies in the independent coding were compared and discussed. In the

majority of cases, they related to some interview aspects that were noticed by one annotator but ignored by the other. In all cases where more than one code could be assigned to the excerpt, the excerpt was rated as belonging to both.

## 3. Results

### 3.1. Participants' Use of English in the Professional Sphere

Sixteen participants described themselves as proficient users of English (levels C1-C2, Common European Framework of Reference for Languages (CEFRL)). As one interviewee put it, "I can say anything I want and I can write anything I want" (His1). Two interviewees identified their level as B2 (CEFRL). One respondent could not determine her level of English because she had always had a feeling that she "under-studied English" because it was not her first foreign language. Based on the participants' self-evaluation, their level of English is sufficient to be used in academic, professional, and personal spheres.

When asked how much they use English in their professional and everyday life, the answers varied: some participants believed that there were "more than enough" English in their life since they taught and wrote mostly in English (His4), while there was an opinion expressed by a historian that there was "not enough English this year" (Soc5). However, all the interviewees admitted that English had become part and parcel of their professional sphere: in their narratives describing the role English plays in their academic life, they used such phrases as "kind of a natural part of my life" (His3), "it's very natural to speak English" (PolSc3) stressing that they could not "perceive myself without English" (PolSc3). A representative from the department of sociology used the metaphor of "living on a little Globe" to describe the feeling English gives to her.

> (1) I have a feeling that I am living on a little globe. Not in a certain country, not in a city, but on the Globe: I either go to a conference, or write someone in English, or someone emails me. And almost every day I have something related to English: I read in English, I write in English. And now I teach in English [ . . . ]. Every day, a little bit, but it's the main [professional] language. (Soc1)

As for the use of English for research writing, the majority of participants (12 interviewees out of 18) named English as the only languages of their recent publications [1]. Three interviewees said that they published mainly in English. Two wrote in English more than in Russian. One participant said that she had "almost an equal number of texts written in English and in Russian." Four participants had papers published in other languages (Spanish, Finnish, and German), but these were mostly translations of their texts written either in English or in Russian. Thus, it can be said that, for the interviewees who participated in the study, English has become either the main or one of the main languages of publication.

### 3.2. Participants' Motivations to Choose the Language of Research Publications

An interviewee working at the department of management explained the reason for choosing English as the language of publication in the following way:

> (2) I don't publish in Russian because I can't do that—I've never done it. (Man1)

This explanation provides a certain characteristic of the current situation in academia: there are scholars, mostly junior and middle-career researchers, who started their professional career when English had already become the lingua franca of academic communication and who had never used L1 as the language for publication. This may be especially true for those scholars who either studied or did their PhD at universities abroad, as the respondent cited above (2).

When discussing the reasons for choosing English as the means of publication, the participants repeatedly referred to institutional requirements, calling them "very harsh" (Soc3) and mentioning that the system "pushes [them] into writing in English" (Soc1).

---

[1] The period when the participants have been writing in only English ranges from four years to 10 years. Mostly, the interviewees referred to the last five years.

From the participants' perspective, these external motivations set very clear preferences toward one language because, according to one historian, "nobody is interested in what you publish in Russian" (His2). The interviewees realized the importance of having publications in English for the university they work for and treated the institutional requirements to publish in a top journal in English as a game where one can either win or lose:

> (3) ... it was clear to us that if you don't play this game, you'll be out. That's why, from the very beginning, I published it in English. (Soc3)

It is worth noting that, in Corcoran's study [38], some participants from Latin America expressed the same ideas describing their perception of English as the main language for publication. They stressed that one had to "play by these rules" or "play something else" (p. 547). It suggests that many multilingual scholars irrespective of the geopolitical region they work in have similar attitudes. Moreover, the interviewees are aware of the fact that national and institutional requirements are similar in many regions of the world. In their narratives, the participants noted that institutional pressure to publish in English was not a specific characteristic of a certain university or a certain country. It was a contemporary academia in general "that makes you write texts in English" (Soc4). Those participants who have experience of working or doing research at different universities abroad, emphasized the uniformity of requirements regarding publications. They mentioned that many universities had "almost the same values, ... the same criteria for publication" (Man3), emphasising that "both here and there you need to publish in English" (PolSc4). The narratives suggest that the institutional requirements, which seem to be the main reason for choosing English as the language of publication by the participants, are not something unique for the university they work for. The value attached to publications in English seem to be equally high in various universities around the world. As one sociologist put it,

> (4) Publications in English are like your dowry, your academic luggage that you can take anywhere, continue in any institution. (Soc2)

What seems to be unique for the university where this study was conducted is the system of economic stimulus that influences the interviewees' decision to publish in English. In their narratives, they mentioned the financial support one gets from the university if the research is published in top journals. One of the historians pointed out a very interesting aspect of financial motivation. It makes you take risks and submit papers to top journals. When such a paper is published, the researcher gains courage to do more research and write better texts (His3).

Sometimes, the participants said that, although the financial motivation was important, it was not the most decisive reason for choosing English as the language of publication, highlighting such personal insights into writing in English as important for their career and for "all kinds of reports" (His4). One historian confessed that she "love[s] writing in English" (His1).

Among other reasons for the preference of English over Russian as the language of publication, the participants named the following:

- absence of readership in the Russian language:

> (5) I consider that, on my topic in Russia, there are no readers in the Russian language. I mean people do not read in Russian. Those who do read, they read English papers. And if the purpose is that someone reads your paper, what's the use of writing in Russian? (Man2)

- the lack of specialized journals in Russian:

> (6) Because there are journals [in English]. I had a colleague who doesn't speak English and we were looking for a journal in Russian to publish, and it was very hard, and we ended up publishing a paper on technological history in a journal of social studies [ ... ]. So it is about variety because there are alternatives in English. (His3)

- English as one of the languages of journals published in Russia:

(7) It is partly connected to the trend I see in Russian journals today. They are also interested in publications in English [ … ], they have strong preferences for texts in English, the leading Russian journals, because it's related to all these metrics, Scopus, Web of Science. (His2)

- the belief that the quality of publications in Russian in general is lower compared to publications in English:

(8) Because there is a belief that all papers of high quality are published in English. I have this feeling. And if you have research that is not of top quality, you publish it in L1—in Russian, in Estonian. (Man3)

In (8), the representative of the department of management refers to the belief widespread in academia (not only among Russian researchers, but also among other L2 researchers) that the quality of publications in English is a priori higher than in other languages.

Another factor that seems to influence participants' choice of the language is the low value of publications generally in Russian. This idea was well formulated by a representative of the department of public administration who explained why she did not write in Russian by the absence of recognition from either university or the research community:

(9) I stopped writing in Russian because the efforts are the same but you get no recognition at all. Either from the university or from the colleagues. (PublAdm)

The attitude toward the fact that publications in Russian have a much lower institutional value, compared to publications in English, varied among the participants. While some treated this situation as natural, others expressed regret and concern about the fact that "nothing is rated in Russian" (Soc1). A negative attitude is well expressed by a historian whose research is related to Russian history.

(10) And I think that it's not fair that publications in Russian are useless publications. They weigh nothing. This is not right because the book I'm writing would have higher public interest here, and it would have a different interest. There, it would be interesting for 20 colleagues. Here, in Russia, it would have a wider readership." (His1)

Such a position is in line with the study of Arnbjörnsdóttir and Ingvarsdóttir [76] who show that scholars from humanities find it natural to publish in the local language on topics connected to local history, literature, or languages.

One political scientist, realizing that in Russia today, there are still many people who cannot read in English, shared with me his intention to write a book in Russian to reach this audience:

(11) I'm thinking of doing that solely because I feel I have to do something for Russia. I can't just sit in my own castle. I must bring ideas to people who don't speak English, to communicate with them. (PolSc3)

In general, although the participants named many reasons for their choice of English as the main language for publication, the most influential ones that were referred to by many participants seem to be institutional policies aimed at increasing the number of publications in English and, hence, at gaining international recognition. Some interviewees regretted that such policies discourage them from publishing in Russian. However, this feeling was not shared by everyone, especially by those who have their personal motivations to publish in English.

### 3.3. Multilingual Scholars' Perception of English as the Language of Publication

When asked in which language it is easier to write professional texts, answers varied significantly. Some participants treated writing in both languages equally difficult. Others confessed that it was easier for them to write in Russian, and still others said that research writing in English was easier.

The interviewees for whom it was equally difficult to write both in English and in Russian named several reasons for that. One of them, according to a sociologist, was that the language was not the most important thing in the writing process.

(12) The main thing is the idea. The writing process itself. The main thing is to formulate, irrespective of the language. (Soc4)

The participants were also talking about problems related to writing in general, such as how difficult it was "to start writing, to 'enter' the language" (His1). Such a position may be attributed to the view of research writing as a skill that needs developing by both L1 and L2 speakers [24,44,45]. However, the fact that some participants seem to treat equally writing in L1 and in English suggests that research writing in English is not considered as something they need to put extra efforts into. It is perceived the same as writing in L1.

Those interviewees whose natural reaction was to write in their L1—Russian—was much easier, gave different reasons. One participant, a political scientist, explained that the main challenge for him was that he had not "started thinking in English." All his ideas were in Russian, and he had to translate them into English (PolSc4). Some scholars were talking about the freedom the native language gave them in the process of writing: they had to "use some clichés" in English while they could "reformulate the same idea in 4–5 different ways" in Russian (PolSc5). A sociologist mentioned that one had to put in more intellectual efforts while writing in English because "you always have to check some words and expressions." (Soc3).

In their narratives, some interviewees implicitly connect the difficulties of writing in English with a native/non-native divide. The participants were talking about the higher variability in the language use of native speakers and about errors that non-native speakers make. Two excerpts below taken from the interviews with sociologists demonstrate these ideas.

(13) Because their [natives'] language is much richer. They know these good expressions. They just have this ease of native speakers. For example, how many epithets can we use—well, 15 positive and about 7 negative. Additionally, natives [can use many more], considering that they are all well-educated, and those who publish in top journals. (Soc3)

(14) In English, some things I can't write without errors, I'm not a native, and I can't think like a native. I can invent five different ways, but I can't give 25 that natives can. (Soc2)

When I asked the interviewee if we really needed 25 ways and if we needed to strive to write native-like, she was sure that we had to.

At the same time one interviewee, a historian, expressed a contrary idea that the limitations imposed by being non-native speaker, e.g., a limited vocabulary, was a benefit rather than an obstacle because you are "not tempted to write something very sophisticated. [ . . . ] You just write" (His4).

The third group of respondents admitted that, for them, it was easier to write research texts in English than in Russian. One of the reasons for that was having more experience in writing research texts in English than in Russian—the practice they developed under the influence of institutional policies of contemporary universities. The participants simply stated the fact that "[W]hen you always write in this language, it gets easier and easier" (Soc1).

Another reason closely connected with the experience of writing in English is that scholars start to better understand English academic writing conventions and, simultaneously, forget those in Russian. In the interviews, the participants mentioned that "[T]he skills are better developed in English" while, in Russian, you had "to reconstruct them":

(15) I haven't written anything for a Russian journal for so long that I have lost this skill in a way. I've lost the idea of what they expect. I think I can do it in Russian, but perhaps it would be a bit hard and it would look somehow unnatural. (His2)

The third reason why writing in English is easier given by many participants is related to professional terminology, which, according to them, "is fixed in English" (Man3). A political scientist mentioned that "in terms of professional terminology, Russian is much more inferior to English. (PolSc1). One sociologist even expressed the idea that since "the academic language in all disciplines is developing first of all in English," the Russian language "is gradually losing its academic vocabulary" (Soc1).

One historian suggested that, since all professional terminology was fixed in English, it influenced not only the ability to write research texts, but also to read, present at conferences and teach. She confessed that sometimes when she had to present in Russian, she "struggled" because "some structures, some frames, some words that are in your head, some lexis, terminology, expressions, they are all in English." (His3). In the interview, I also asked the participants if they ever had the situation when they had to write the same content in two languages, such as a grant proposal. I was interested in which language they chose to write the first text. The answers further revealed the respondents' practices and preferences in the languages for research writing. When such situations happened, many participants reported that they had used English first. It is interesting that, sometimes when the English text was written, they asked either "the colleagues whose English was not that good" (His2) or students (Soc3) to translate it into Russian. As one interviewee, a representative from the department of management, explained:

(16) I write in English first because I know how to put it in English better. (Man3)

Another explanation given by the participants for choosing to write in English first was that it was "faster this way" (PolSc2). The fact that several participants described the practice of writing the text in English first, confirms the ideas expressed in the interviews that writing research texts in English had become easier for them than writing in L1. This may be regarded as further evidence that English has become the "L1" for research writing for many multilingual scholars. This conclusion, however, does not refer to every multilingual speaker. There are still scholars, even among those who are successful in their international research career, who have to use English under the influence of different social conditions they work in not because it is their choice to use English. One political scientist who participated in the interview expressed a very negative attitude to the necessity to write and teach in English:

(17) I don't want to write in English, really. I want to write in Russian. And I don't want to read lectures in English, I want to do it in Russian, I can do it better in Russian. But I have to. (PolSc4)

Although such an attitude was an exception since the participants were generally rather positive in their attitudes toward publishing in English and using English in general, it suggests that the situation in the contemporary academia outside Anglophone countries is not homogeneous.

### 3.4. Attitudes to and Perception of the Publication Process in English

To elicit the discussion about the publication process in English, I directly asked the participants, if, from their perspective, it was difficult to publish in English. Surprisingly, only one scholar, a historian, described this process as such:

(18) It's difficult because you need to understand where you want to publish, who you write for. There's a feeling that you are playing according to other people's rules. And this is a hard feeling to know that these rules are not set by you. There's even this expression—gatekeepers. And the longer I live in my professional sphere, the better I understand that these other people can interfere. (His1)

Other participants were not so categorical in their assessments. It is interesting that, in their narratives, they associated difficulties in publication not with the language itself, but rather with such factors as journal qualities, the quality of the argument, and the research

process in general. Talking about journals, they mentioned that "the higher you climb in the journal ratings, the harder it gets" to publish your research (Soc1). The argument quality as the most influential factor in the reviewers' and editors' decision was named by several interviewees. In the excerpt below, a sociologist gives a detailed description of the difficulties she usually faces in the process of research writing:

(19) The difficulty lies in the construction of your argument, in its justification. The difficulty is in finding the novelty of your research, in making your text interesting and appealing. [ . . . ] Personally, for me, the problem is not in the language you use, the problem is in finding time to dive into the text, to read loads of studies related to your topic. (Soc5)

Similar ideas were expressed by the interviewees who did not see any difference in writing research in English and in Russian. From their point of view, the writing process is equally difficult in both languages because "it's difficult to publish in general" (Soc2) and "it seems that this process is not simple." (Soc4). Their idea was that it was not the language that caused difficulties because the publication process was hard irrespective of the language you publish your research in. Some participants, although, believed that the publication process in Russian was easier and faster. For example, a sociologist reflecting on the idea that Russian scholars prefer to submit high-quality papers to English-medium journals and, consequently, Russian journals lack qualitative publications, said that good papers in Russian were published much faster.

(20) Of course, our papers in Russian are accepted for the next issue. I mean, when we submit a paper in Russian [ . . . ] we publish it within a month. In English, it's a totally different story. (Soc3)

However, this opinion was not shared by every participant. Another sociologist expressed the idea that currently top journals in Russia also have high requirements and, hence, "it's the same, to publish in Russian and in English [ . . . ] the culture is the same everywhere." (Soc5).

In the discussion of language-related issues in the publication process in English, the participants confessed that they had some problems in the use of English. Mostly, they referred to lexis (limited vocabulary) and grammar (the use of articles). However, when asked if language problems had ever been the main reason for rejection, none of the participants answered affirmatively. They said that the language might have been one of the reasons, but, mostly, the editors' and reviewers' decisions were not based on linguistic issues. The participants either referred to the comments on the language as "always a minor point" (Soc3) or explained that the comments to improve the language mean "the language in the widest meaning: not only articles, but logic, arguments" (Soc1). In general, they noted that the reviewers' comments were "more about the quality of your texts, the quality of your arguments, [and] not about your language." (His3). It is interesting that, sometimes, after saying that their papers had never been rejected because of English, the interviewees started telling the stories of other people who had this experience. A historian mentioned "a colleague in Berlin whose level of written English was not very good" and who was told that the language in the paper was poor (His2). A representative from the department of management spoke about "some other people from the laboratory" who had comments on the language (Man2). One interviewee, for example, learned that the paper could be rejected because of the language from her supervisor.

(21) I was told about this. I heard stories, well, my supervisor, he was a journal editor, he told me, he had such cases. (His1)

The excerpts above suggest that some beliefs about language difficulties experienced by L2 speakers may be based on some 'lore' or 'received wisdom' [12] that "do not always align with the findings of empirical research or engage with the theoretical discussions advanced in much scholarship on multilingual scholars writing for publication" (p. 2).

The interviews revealed that some participants always had their texts proofread, others used to but abandoned it at some point, and still others tried proofreading once and

were not satisfied with the results. There were also interviewees who had never given their texts for proofreading.

Those who sent texts to proof-readers did it for various reasons. Some interviewees believed that it was not possible to publish in top journals if the texts are not proofread or edited professionally. They said that "it's useless if you want to publish in Q1-Q2 journals" (PublAdm) noting that "the imperfection of the language" might lead to some "loss of meaning" (Man3). In one interview, a scholar from the department of management referred not to the errors non-native speakers make, but to the belief that native speakers' texts are a priori written better and are easier to read and comprehend. It is interesting that this belief is imposed by the gatekeeper—the journal editor:

> (22) Now an American editor insists on editing, he says that [you have to do it] if you want your paper to be popular, to be read and cited. Your text is comprehensive. You have no clear mistakes. But imagine that there are two abstracts and introductions [ . . . ]. If someone takes two papers on a similar topic, one is yours and another one written by a native speaker, they would prefer the paper written by the native because it would read easier. (Man2)

Another belief spread among the participants that can be viewed as a reason for proofreading is that the level of L2 speakers of English would never be high enough and that there would always be mistakes in texts written by them. The participants said that there are a number of errors they always make such as with articles and prepositions. One scholar even expressed an opinion that "you have to accept it that you will never learn to use them [articles] correctly (Man2). The interviewees confessed that they "absorbed this idea" that they "cannot make it 100% correct" (Man3). One interviewee, a historian, used the expression "impassable barrier" to explain that multilingual speakers would never be able to write like native speakers.

> (23) Because it's an impassable barrier. First, we'll never be able to insert the correct article. Second, there are many things that can be better expressed by native speakers because, when you use the same words, you render different nuances [and] different shades of meaning. They can find something better [and] something more suitable. (His4)

It is interesting that the same interviewee, who assessed his level of English as C2 and who always gave his texts for proofreading, recognized that the proofreading experience might be different and it could not solve all the problems. He told a story when he received reviewer's advice to have the text proofread after it had been edited by some company that provided professional editing services. At the end of the story, he called "funny" the situation "when you give the texts edited by one native speaker to another native speaker, and they make another 25,000 corrections." (His4).

Negative experience with proofreading was mentioned by a sociologist who stopped practicing proofreading for almost the same reason. They received many comments on the language and advice to have the text proofread by a native speaker after they had done it (Soc4).

Despite descriptions of some negative experience, in general, the interviewees treated the proofreading process as highly valuable. Some of them like reading texts after proofreading "because they are so beautiful" (PublAdm). Some participants said that they analyse every mistake they made and, thus, work on their English (His3) and added that they are happy if they see that they are making fewer mistakes. One interviewee treated the proofreading process as "a perfect chance to look at the text again and to improve something" (Man3). A very positive attitude toward proofreading was expressed by a historian who always had her texts proofread.

> (24) I think it's natural when the text is proofread by a native speaker, in the language of that native speaker, be it English, or Russian, or Finnish. Because I realize how important it is to formulate well. [ . . . ] And speaking about money you have to pay [laughter], well, I'm used to that somehow." (His3)

At the same time, another historian confessed that he ignored advice to proofread the texts mostly because, from his point of view, it was not worth the money you had to pay.

(25) This might sound like a snobbish approach, but I am ready to sacrifice some insignificant improvements you get if you have to pay for that. (His2)

Discussing language-related issues in the publication process and participants' attitudes to proofreading, I asked the interviewees if they felt any unfairness to the situation when L2 speakers need to take some extra effort, such as in proofreading. I also asked if they felt any discrimination because their L1 was not English. Only one historian said that she felt underprivileged when compared to L1 speakers.

(26) Yes, I feel this discrimination. And it reveals, first of all, in that I always need to find people who would correct my language. (His1)

In other narratives, the attitudes were more neutral. Some interviewees described their feelings as "okay" because they "learned how to survive in this world" (Soc3). Some said that they were "absolutely fine about it" (Man3). One interviewee, which is a historian, described the contemporary academia as "a highly competitive market" and expressed his belief that, if the language of the paper was comprehensible, the thesis presented in the paper "would be far more important than linguistic errors" (His2). Another participant, a sociologist, called the publication process "a conveyor" with many different people being involved in the production.

(27) Well, I treat it all as a conveyor. And my attitude to that is very simple [ . . . ]. People have different English skills. Editors, reviewers, writers. And I take comments related to the form of my paper—language, structure [ . . . ] very easily. Very calmly." (Soc3)

Two excerpts below summarise the general attitudes expressed by many interviewees who participated in the study.

(28) There's no discrimination. These are the rules of the game we are all playing. (PublAdm)

(29) What's your attitude towards it?

- Very calm.
- It doesn't hurt you?
- No, not at all.
- And you don't feel any discrimination?
- No. Moreover, you understand that it totally depends on the reviewer [ . . . ]. One reviewer, a non-native, would say that everything is okay, while another one wants a good style, and they ask to polish the text.
- So, you treat it as a normal, natural process?
- Yes, normally. And we constantly discuss with colleagues which journal they submitted, where they sent their texts for proofreading, and what the results were. (Soc5).

In general, it can be said that, although the participants' attitudes to the publication process in English vary, the majority of them seem to treat it as a natural process. The majority of the participants do not seem to feel any unfairness, any bias, or discrimination that was witnessed in some studies of multilingual scholars' research writing practices.

## 4. Discussion

The interview analysis has shown that, for the scholars who participated in the study, English has become an integral part of their professional life. It has also become either the only or the main language of publication. This situation is not unique. Similar results have been described in studies conducted in different geo-linguistic contexts [21,22,24,26]. This suggests that the participants of the study may be called typical representatives of multilingual scholars outside the Anglophone centre who are successful in their research career. By uncovering the factors that motivate scholars to use English and to find out

their perception of English as an academic lingua franca, this study contributes to the on-going discussion of the position of multilingual scholars working outside Anglophone countries in contemporary academia, which is "essentially English as a lingua franca setting" [4], (p. 6).

The findings of the study provide insights into the social and personal factors that influence Russian scholars' choice of languages. One of the factors is scholars' educational and research experience. The interviews revealed that some of the participants either had never published in Russian or had not published in Russian for a long time and, hence, had forgotten how to write scholarly texts in Russian. This might be explained by the age of the participants since the majority of them started their academic careers when English was widely used in academia. Some of them received degrees in English. It means that, in the global academia today, there is a new generation of scholars who perceive English as the L1 of their professional communication and, hence, are very likely not to treat it as an additional burden they have to carry.

The fact that some multilingual scholars have never written research texts in L1 implies that social contexts the scholars work in create conditions by either allowing them to publish only in English or stimulating them to do that. In their narratives, the participants referred to the institutional policies and requirements that seem to be the main reason why they choose to publish texts in English. This is in alignment with other studies of multilingual scholars' practices that describe different institutional policies aimed at increasing the number of publications in top-ranked journals [14], which suggests that the requirements to publish in top-ranked journals, although imposed by a certain university, might be treated as unwritten regulations imposed by contemporary academia.

It is worth noting that the interviewees realize that the requirements to publish in English is not a specific feature of the university they work in. This may explain why many interviewees positively treat the requirements. They view publications in English as their research "luggage" that allows them to be competitive in the international research arena. This conclusion is not surprising. If the requirements in terms of a scholar's research output are similar in the majority of universities that compete for high positions in international rankings, then having publications in top journals in English is scholars' assets that they value very highly.

Besides the requirements that seem universal, there are institutional policies that do impact the participants' research writing practices. The academic merit bonus system that the interviewees recurrently referred to in the narratives is an important factor encouraging publishing in English and discouraging publishing in Russian. Lower quality of publications in Russian, lack of publication venues in the local language, and lack of readership, mentioned by the interviewees, may be regarded as the consequences of such policies. A low value of publications in local languages, as previous studies have shown, is a common trend in many non-Anglophone countries [5,18,25–27].

However, the fact that this trend is enhanced by the university policies, raises concerns in some participants' narratives. The successful scholars who participated in the study realize that they deprive local research community of receiving new knowledge by "sitting in their own castle" and disseminating knowledge only in English. This was especially evident in the interview with the scholars whose research topics are related to the Russian local context (language, culture, history). Similar observations were made by Arnbjörnsdóttir and Ingvarsdóttir [76]. In the interviews, some participants expressed their desire to write some research in Russian in order to do something for their country. Similar intentions are described by Duszak and Lewkowicz [26] in their study of scholars from Poland who viewed research publications in L1 as a way "to disseminate their field of study among Poles and 'popularize' their subject" (p. 115). On the other hand, there was an opinion expressed by one of the participants that today everyone who needs this knowledge is able to read in English. Based on the interview analysis, it may be concluded that, although the participants seem to value publications in English, they feel somehow conflicted about their publication practices.

The discussion of the participants' perception of English and the challenges they face in the process of writing and publishing in English showed that, while the interviewees' views were not homogeneous, there seemed to be an agreement among the participants in their positive attitude toward the use of English in their professional sphere. However, the perception of the research writing in English varied. There was an opinion that writing in English was difficult and if there were a choice, one interviewee would choose to write in Russian. It suggests that, in Russian universities, we have scholars who struggle to write in English. However, to discover the exact proportion, we need to conduct other larger-scale studies. In this sample, the majority of participants did not view research writing in English as something they needed to put extra efforts into. They either considered writing in both languages (English and Russian), which is problematic, stressing that research writing was a difficult process in general. Furthermore, they confessed that research writing in English was easier for them than in Russian. This echoes results of other studies of multilingual scholars' writing practices [46].

Describing their writing practices, many participants said that, when they needed to write the same content in two languages, they chose English to be the language of the first texts and then translated the English version into Russian, either by themselves or they asked someone else to translate. Such an attitude to English gives further evidence to the fact that a new generation of multilingual scholars have research writing skills that are better developed than in their L1. The interviews revealed that this could be related, on the one hand, to the lack of experience in L1 research writing, and, on the other hand, to wider general experience in the use of English in the professional sphere (reading, teaching, and communicating).

Discussing the publication process in English, some interviewees were more positive toward it than were the others. Those who did not see many differences between publishing in English and in Russian treated the publication process as difficult. Those who referred to the difficulties they faced in the process of publication in English, were talking about the content of the paper, the arguments, and the journal requirements. They were not talking about difficulties related to the language used. This suggests that multilingual scholars might have the same challenges L1 speakers have [47]. Of course, it is not possible to generalize that all multilingual scholars working in Russia treat the publication process in English rather positively. The attitude toward it expressed by one participant who felt especially strongly about publishing in English compared it to playing a game "according to other people's rules," which suggests that it is a complicated and probably painful process for some scholars. However, the attitudes of the majority of the participants imply that the publication process in English is becoming or has become a normal part of their professional life, and they treat it as something natural.

Although realizing that language was not the main thing that influenced reviewers' and editors' decisions about manuscripts, the participants seemed to have a clear orientation toward "native-like" writing. They want their texts to be errorless. Hence, they use proofreading services—the norm imposed by many journals. In general, the narratives revealed how some "lores" (in the terminology of Curry and Lillis [12] are formed in an academic community).

As reported in the findings, the proofreading is used by many participants. In general, it may be said that participants' explanations of why they perform proofreading despite the fact that they view their level of English high enough, suggest that there is still a belief that non-native speakers of English should always have their scholarly texts in English proofread. The interviews showed that the practice of proofreading is treated differently by the participants. Many of them seem to have a positive attitude toward proofreading, perceiving it as a means to improve their English and develop their writing abilities. On the other hand, some participants described a negative experience of proofreading. Some interviewees had their papers published without being proofread. The most important conclusion about proofreading practices may be that the participants, irrespective of whether they perform proofreading or not, perceived it as a natural process and did not

seem to treat it as an extra burden they had to carry because their L1 was not English. Discussing the financial costs associated with proofreading, the interviewees confessed that, even if they paid for the services (although the university provides consultations with proof-readers), they knew that, when the paper was published, they would be rewarded financially by the university. Such an approach illustrates that the university's policies apart from shaping research writing practices in a certain language (English), also help in forming a positive attitude toward it.

**5. Conclusions**

In general, the interview analysis suggests that the majority of the scholars who participated in the study, who are more experienced rather than novice writers, do not perceive themselves disadvantaged when compared to L1 speakers of English. They mostly described their experience of writing and publishing research in English in a positive way, highlighting the benefits they receive rather than the burdens they carry. They were conscious of all possibilities that English provides in disseminating their research globally and perceived themselves as full participants of international research communities. This echoes some other studies questioning the myth of the linguistic injustice [40,47,52,77].

This conclusion should be viewed in light of the study's limitations. It is applicable only for the study participants who work at a leading university in Russia and who are successful in their research career. It cannot be generalized to apply to the whole academic community in Russia because universities' policies and requirements as well as resources they invest to support scholars in their research vary significantly across the country. It also needs mentioning that the participants are rather experienced writers who have several papers in English published in international journals, which influences their perceptions of writing and publishing in English. However, the study sheds some light on the situation in the leading universities in Russia that aim to be ranked high in international university rankings and to build an international reputation. It also gives insights into the perception of research writing and publishing practices of established multilingual scholars.

To conclude, I would like to allude to the metaphor of the game used in the title of the paper. This metaphor was found in three narratives. Two interviewees referred to the game where the multilingual scholars were playing by trying to get their research published in a positive way, implying that one needed to learn the rules and win. One scholar, however, confessed that it was difficult for a multilingual speaker to win the game since other people set the rules. These two approaches describe rather accurately participants' general attitude to English, research writing in English, and a publication process in English. While some scholars struggle with English, others treat it positively, viewing professional communication in English as a very natural part of their life. However, irrespective of the attitude to English, the scholars seem to see more benefits than costs in writing and publishing in English. Such perceptions, as the study showed, are formed not only by institutional policies and requirements, but also by policies at the national and at the global level.

Taking into account the similarity of policies regarding publication in English in different universities around the globe, it is very unlikely that these policies will change in the near future. What might be changed are the policies discouraging scholars from having publications in local languages. A wide public discussion of the concerns expressed by the scholarly community may bring some results. While this manuscript was under revision, the university where the research was conducted launched a new initiative that aims at supporting and advancing the Russian language as a language of science. It announced the annual competition for the best academic paper and popular science project published in Russian. This is an example of a small initiative compared to the systematic policies to increase publications in English, but, in the long run, similar initiatives can hopefully make some differences. They may solve the conflicts that some successful multilingual scholars have about their publication practices.

**Funding:** This research received no external funding.

**Institutional Review Board Statement:** Not applicable.

**Informed Consent Statement:** Informed consent was obtained from all subjects involved in the study.

**Data Availability Statement:** Not applicable.

**Acknowledgments:** I would like to thank the scholars who participated in the study for the meaningful conversations we had, for their help, and suggestions.

**Conflicts of Interest:** The author declares no conflict of interest.

**Appendix A**

**Interview Protocol**

Background questions

- How old are you?
- What is your education?
- What languages do you speak? (please, estimate the level)
- Do you have any experience studying or working abroad? Give details.
- How many publications do you have?
- What is the approximate number of publications in Russian, in English, in other languages?

Questions about disciplinary affiliation and research methodology

- What is the main sphere of your research interest?
- Can you name the discipline(s) your research belongs to? (Probes: If we use subject areas, e.g., in Scopus, what area it would be? Is it one discipline or a multidisciplinary field? If more than one discipline is named: in which discipline do you feel most confident as a writer? Choose the discipline you know the best that we could discuss).
- Have you always done your research in [discipline]?

Questions about use of languages in a professional sphere

- How much do you use English in your professional sphere (teaching, doing research, writing)? Why?
- Is it difficult to teach in English? What are the main difficulties (if any) (probe: in which language is it easier to teach?)
- In which language(s) do you read professional texts? (probe: do you read in Russian? Why?/Why not?)

Questions about the language of publication

- In what language do you publish your research? Why? (probe: why do you choose English/Russian? why don't you publish in Russian?)
- Is it difficult to write texts in English? What are the difficulties? (probe: in which language is it easier to write professional texts? why?)
- If you were to write the same text (content) in Russian and in English, in which language would you write the first text?
- How important are linguistic issues?

Questions about the publication process

- Is it difficult to publish in English?
- How important are the linguistic issues in the process of publication?
- Have linguistic issues ever been the main reason for rejection?
- Do you have your texts proofread? Why?
- What is your attitude about having your texts proofread?
- Talking about the publication process in English, do you feel disadvantaged when compared to native speakers of English? (Explain why).

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
