# Peer review of "“There’s No Discrimination, These Are Just the Rules of the Game”: Russian Scholars’ Perception of the Research Writing and Publication Process in English"

_publications, doi:10.3390/publications9010008_

Round 1

Reviewer 1 Report

The paper is well written, well researched, and well situated within a body of research in English for Academic Purposes. My main concern is that the author does not do enough to distinguish the present study from previous ones, other than the fact that it looks at Russian scholars rather than those from elsewhere. While that is certainly a gap that needs to be filled, I believe the author can make a stronger case for why Russian scholars are of particular interest, given the growth in internationalization and status that their universities are experiencing. The information is there; it simply needs to be developed further near the beginning of the paper. Furthermore, the connections between the findings of this study and previous ones could be made stronger in the discussion section. See my detailed comments below:

Lines 22-30: First, the author needs to remove the first paragraph of the Introduction, which appears to be unintentionally copied from the journal guidelines and instead begin with the second paragraph, which starts with “One of the key […]”

Line 35: “high reputation” should be “strong reputation” (collocation)

Line 38: “within and without” should be “within and outside of” (more idiomatic)

Lines 58-63: Format block quote so that it is offset from the rest of the paragraph.

Line 80: When first introducing the “game” metaphor, the author may want to refer to the work of Chris Casanave, specifically Writing Games: Multicultural Case Studies of Academic Literacy Practices in Higher Education.

Line 108-109: Casanave is misspelled and a short quote or paraphrase from her work would add clarity as to why “writing for publication never gets easier.”

Line 141: I would suggest using a word other than “appealing” to describe the motivation for this study; it sounds like it is simply a matter of convenience. I would suggest perhaps “necessary” or “imperative” instead.

Lines 148-154: The gap the present study fills could be made clearer with a bit of rewording: “While these studies were conducted in various geographic locations, none of them examined Russian scholars’ perceptions of research writing and their publication practices. This is of vital interest since Russian universities have put forth considerable effort to be ranked highly among international universities, which has resulted in […]” 

Line 313: Change “16” to “Sixteen” (convention)

Line 322-325: Re-format to make excerpts from interviews more distinct from the rest of the text, and then repeat with all that follow.

Line 349-350: Rephrase the opening of the sub-section to make a statement before introducing the excerpt: “One participant explained the reason for choosing English as the language of publication:

            (6) I don’t publish in Russian […]”

Line 454: Consider adding a brief summary of the entire sub-section before moving on to 3.3

Lines 456-457: Revise the sentence for clarity: “When asked in which language it is easier to write professional texts, answers varied significantly: […]”

Line 458: add “and” between “Russian,” and “still” for better flow.

Line 503: This is an important point and merits further consideration. Why do they have more experience? How might this raise issues of fairness or undue burdens?

Line 563: Change “as a hard one” to “as such” (more formal)

Line 590: Change “though” to “however”

Line 742: As noted earlier, it would be useful to have a brief summary of the excepts presented before moving on to the Discussion section.

Lines 746-749: Here is another opportunity to elaborate on the significance of the present study: Aside from addressing a different geopolitical context, what does this study contribute to ongoing conversations about publishing in English as an academic lingua franca?

Line 752: Does the author mean “write scholarly texts in Russian” ?

Line 788-791: Elaborate on the studies by Arnbjörnsdóttir and Ingvarsdóttir (2018) and Duszak and Lewkowicz  (2008): what exactly did they find that relates to the present study?

Line 796-797: Revise: “[…] publishing in English showed that, while the interviews were not homogenous, there seemed to be agreement among […]”

Line 800: Does the author mean “one interviewee” rather than “the interviewee” (which makes it sound more widespread among participants)? If it is the latter, then I would suggest “most interviewees” instead.

Line 802: Revise: “to discover the exact proportion, we need to […]”

Line 813-814: Revise: “[…] have research writing skills that are better developed in English than in their L1.”

Lines 828-829: Elaborate briefly on exactly how “the scholars’ views are different.”

Lines 832-834: This point seems contradictory: If the participants realized that “language was not the main thing”, then why did “they want their texts to be errorless”? Consider rephrasing or explaining it further.

Line 838: Does the author mean “[…] because of the quality of their English influenced one interviewee’s practice […]” ?

Line 844: Does the author mean: “non-native speakers of English should always have their scholarly writing in English proofread” (because I would think there are many lower-stakes instances of English use that don’t require or would not permit proofreading)?

Line 858: I would suggest starting a new section here titled “Conclusion”

Line 866: “study’s limitations”

Line 877: Change “contrary” to “however”

Line 884-885: The author could weigh in stronger on this point at the end of the paper. Yes, it is true that policies at all levels form these perceptions, but policies can be changed, and may be in positions to change them, at least locally. Should we? How so or why not?

I hope these recommendation will be helpful as the author prepares this manuscript for publication.

Reviewer 2 Report

Overview

This piece has the potential to be in important contribution to the growing body of work on multilingual scholars' experiences with and attitudes towards research writing for publication. In particular, it highlights the experiences of a particular group of high-achieving, high English proficiency, mid-career scholars from across a range of social science disciplines. While the article adds to the field in a meaningful way, some substantial changes should be made prior to publication.

Critique

My first point is that the characteristics of the group under investigation is under-emphasized in the introduction and discussion sections. Discussion should suggest how these findings add to the literature on more experienced - rather than novice - social science scholars in this geolinguistic locale. Interestingly, as you mention, there is evidence that scholars continue to feel conflicted about their publication practices, especially when they are tasked with solving local / regional problems.

Next, the findings section needs to be completely revamped, using more summarizing of qualitative findings rather than clumping them all together. It would also be nice to know at least the disciplinary background of scholars who are quoted in the section. Both of these suggestions - which MUST be attended to robustly - are in service of "readability" and will result in a more coherent, easily-digestible piece that will be read by a wider audience.

My final area of critique is that I would like to see the institutional writing support angle emphasized more. For many of us who support plurilingual EAL scholars' research writing, it is always disappointing to see such uneven (and at times non-existent) writing support at research-intensive universities, especially given the increasing expectations for publishing in high impact journals. For me, this is the most important element, rather than, for example, whether "linguistic bias" is indeed a (perceived) reality. 

Some additional literature to include might be Flowerdew (2019) and/or Corcoran (2019) and/or Hultgren (2019; 2020) work on perceived linguistic bias. 

Notes on style

Replace the word "out-dated" in the abstract. The suggestion of systemic bias in global knowledge production - whether perceived or real - is most certainly not "outdated", regardless of whether your findings point to a particular orientation / attitude among mid-career Russian social scientists. 

Delete the 1st paragraph in the introduction. Not sure what it is doing there - seems to be instructions to authors.

p. 3 Hyland (2016) is a position piece rather than a "study" so perhaps the preceding sentence could read "several scholars..." rather than "studies"

p. 4 Many scholars doing research in multilingual writing refer to "linguistic repertoire" in a way that highlights the translingual / plurilingual interaction between languages when producing texts. Perhaps be clear that you are referring to linguistic repertoire with respect to language choice.

In the methods section, I would prefer to see passive rather than active voice used.

p. 6 - "variability" or "validity"?

p. 17 temper claims where necessary, using modal verbs to highlight the particular characteristics of the scholars under investigation

Final comments

This piece can add to the growing body of work surrounding the incentives / disincentives for global scholars to publish their work in English vs. other languages, and how these pressures are taken up by scholars across geolinguistic locales. I hope to see this article make it to "print" in the near future - keep up the good work!

Reviewer 3 Report

This is a very well written and thoroughly researched article. The references are updated and diverse. Evidence is provided through sound methodology which included double verification of interview coding system.

Please correct the following mistakes:

ll. 22-30 - This paragraph belongs to the "Recommendations for Authors" and was left inadvertently.

l. 108 Casanava / Casanave

l. 173 Replace the ?, because this is not a direct question.

l. 272 Supress "years"

ll. 407-411 + 437 Surely the numbers of the examples are missing? Please replace [Number] with (19), (20), etc.

l. 449 Replace "o" with "to"

l. 784 "The successful scholar participated in this study realize (...)" - there's a "who" missing, I think.

Author Response

I would like to thank the Reviewer for their valuable comments that helped me to revise the texts. I respond to the comments point-by-point below. The original comments appear in bold font, and my responses follow in italics. 

Please correct the following mistakes:

  1. 22-30 - This paragraph belongs to the "Recommendations for Authors" and was left inadvertently.

I have done it, thank you

  1. 108 Casanava / Casanave

I have corrected the name – thank you for pointing to that

  1. 173 Replace the ?, because this is not a direct question.

I have done it, thank you

  1. 272 Supress "years"

I have done it, thank you

  1. 407-411 + 437 Surely the numbers of the examples are missing? Please replace [Number] with (19), (20), etc.

Thank you for pointing to that. I have done it, thank you, but since the Results section has been significantly changed following the recommendations of reviewer 2, the examples you mention are not in the text any more.

  1. 449 Replace "o" with "to"

I have done it, thank you

  1. 784 "The successful scholar participated in this study realize (...)" - there's a "who" missing, I think.

I have added “who” - thank you for pointing to it.

Round 2

Reviewer 1 Report

Thank you for making the requested revisions to your manuscript and for clarifying certain points. I think this paper will make a fine contribution to the field of EAP in its present form.

Reviewer 2 Report

This piece is much improved - well done! I suggest revising the "Results" section once again in order to streamline the salient participant quotes - all other findings should be summarized via paraphrasing. Otherwise, my only other qualm is that the text could use another round of copyediting (e.g., use of definite vs. indefinite articles); however this is a minor issue that does not impact intelligibility.

Reviewer 3 Report

This revised version has considerably improved an already high-quality research article.

Please revise the 1st sentence under 5. Conclusions – I think there’s a “who” missing.